# Large Deviations for Continuous Time Random Walks

**DOI:** 10.3390/e22060697

**Published:** 2020-06-22

**Authors:** Wanli Wang, Eli Barkai, Stanislav Burov

**Affiliations:** 1Department of Physics, Bar-Ilan University, Ramat-Gan 52900, Israel; Eli.Barkai@biu.ac.il; 2Institute of Nanotechnology and Advanced Materials, Bar-Ilan University, Ramat-Gan 52900, Israel

**Keywords:** large deviations, diffusing diffusivity, saddle point approximation, continuous time random walk, renewal process

## Abstract

Recently observation of random walks in complex environments like the cell and other glassy systems revealed that the spreading of particles, at its tails, follows a spatial exponential decay instead of the canonical Gaussian. We use the widely applicable continuous time random walk model and obtain the large deviation description of the propagator. Under mild conditions that the microscopic jump lengths distribution is decaying exponentially or faster i.e., Lévy like power law distributed jump lengths are excluded, and that the distribution of the waiting times is analytical for short waiting times, the spreading of particles follows an exponential decay at large distances, with a logarithmic correction. Here we show how anti-bunching of jump events reduces the effect, while bunching and intermittency enhances it. We employ exact solutions of the continuous time random walk model to test the large deviation theory.

## 1. Introduction

Following the erratic motion of pollen under his microscope Robert Brown discovered what is called today: Brownian motion. This phenomenon was modeled by Einstein and others, with the random walk theory while the mathematical description of Brownian motion, i.e., the Wiener process [1], was quickly established soon after. Two ingredients of this widely observed dynamics are that the mean square displacement increases linearly with time and that the spreading of the packet of particles all starting on a common origin is Gaussian [2,3]. The latter is the physical manifestation of the central limit theorem. However, in Reference [4] Chaudhuri, Berthier, and Kob, discovered what might turn out to be a no less universal feature of random walks. Analyzing experimental and numerical data, e.g., the motion of particles in glass forming systems, they revealed an exponential decay of the packet of spreading particles; see related works in References [4,5,6,7,8,9,10,11,12,13,14,15,16,17,18,19,20,21,22,23,24,25]. Soon after this, a similar behavior described by the Laplace distribution, was found for many other systems, including for molecules in the cell environment [7,26,27]. In [4] it was suggested that a specific type of continuous time random walk (CTRW) model could explain the physics of the observed behavior. Two of us have recently shown that the phenomenon is indeed universal as it holds in general [28] and under very mild conditions (see below). The goal of this paper is to produce further evidence for the phenomenon, present exact solutions of the model and compare it with the new theory. We also present a more detailed analysis of the model covering cases not discussed previously.

The observed behavior, is related to the way experimentalists record the erratic paths. Following many trajectories one may find universal features of the motion in at least two different ways. In some situations the measurement time is very long such that many jumps occurred and consequently Gaussian statistics will take place. However, in practice experiments are not conducted for an infinite time. In fact in many experiments, the motion follows some trapping and then released motion, i.e., a hop and then wait dynamics, as modeled by the CTRW. In these experiments one follows many paths, e.g., many single molecules, however typically each trajectory is recorded separately. When the averaged number of jumps recorded under the microscope (or in simulations) within the observation time (0,t) is not large, one would expect naively that this would imply the non-existence of universal statistical laws. However a second limit, i.e., the limit of a large number of jumps when |x|/t→∞, is important, which is large positions (and finite times). In this case, we will promote the idea that exponential spreading is the rule. Thus the observation of either Gaussian or exponential spreading depends on the time scale and length scale of the motion and in statistical sense the temporal and spatial extent of the field of view. The idea is that by observing the density of packet of particles in its tails (large |x|) we are in fact considering trajectories where the number of jumps is large, compared to the mean. We advocate that universal laws, that demand a large number of steps, can be found for large displacements (and fixed time) or for large time as usual. Theoretically since we have two parameters that can be made large, i.e., t/〈τ〉 or |x|/σ we may obtain more than one limiting law. Here 〈τ〉 is the average time between jumps, and σ is the variance of jumps lengths. The motion is unbiased hence the finite timemean jump size is zero. We will consider, among other things the case, when both t/〈τ〉 and |x|/σ are large and their ratio is finite.

The remainder of the manuscript is organized as follows. In Section 2 and Section 3, after presenting the CTRW model, we consider the far tails of the distribution of the number of renewals and the position of the random walker, where the waiting times are drawn from the exponential and the Erlang distributions, respectively. The bunching effect is investigated in Section 4 using the sum of two exponential probability density functions (PDF). Finally, we conclude with a discussion.

## 2. Appetizer for Exponential Tails

We consider the well known CTRW model [2,29,30]. This model describes a wait and then jump process. A particle in dimension one, starts on the origin at time t=0. We draw a waiting time from the PDF ψ(τ) and then after the wait, the particle will perform a jump whose length, χ, is drawn from f(χ). The process is then renewed. In our case the waiting times and the jump lengths are not correlated. The position of the particle at time *t* is $x\left(t\right)=\sum_{i=1}^{n}\chi_{i}$, and here *n* is the random number of jumps within the time of observation (0,t). The focus of this manuscript is the PDF of finding the particle on *x* at time *t* which is denoted P(x,t). In particular the large *x* limit of this distribution is of interest. We focus on non-biased CTRWs, and then if the mean waiting times and the variance of jump lengths are finite the density P(x,t) will converge to a Gaussian as expected from the central limit theorem. This limit theorem is valid for t→∞ and $\left|x\right|\propto t^{1/2}$, while here we are interested in finite time effects and the large |x| limit, to be defined more precisely below. Generally, the solution of the model is given by
(1)$$P\left(x,t\right)=\sum_{n=0}^{\infty }Q_{t}\left(n\right)P\left(x|n\right).$$
Here the sum is over the possible outcomes of the number of jumps in the process, $Q_{t}\left(n\right)$ is the probability of attaining *n* jumps [31], while P(x|n) is the probability of finding the particle on *x* conditioned it made *n* jumps. In Laplace space [28],
(2)$${\hat{Q}}_{s}\left(n\right)=\int_{0}^{\infty }Q_{t}\left(n\right)\exp\left(-st\right)dt=\frac{1-\hat{\psi }\left(s\right)}{s}{\hat{\psi }}^{n}\left(s\right),$$
where *s* is the Laplace pair of *t*. In particular, when n=0, $Q_{t}\left(n=0\right)=\int_{t}^{\infty }\psi \left(\tau \right)d\tau$ is called the survival probability.

Generally finding $Q_{t}\left(n\right)$ and P(x|n) is non trivial. Here the goal is to consider special choices of jump length distributions f(χ) and waiting time PDFs ψ(τ) which allow us to express the solution as an infinite sum over *n* explicitly, i.e., without restoring to inverse Fourier and Laplace transforms and/or numerical simulations. This allows us to compare between exact solutions of the problem, and the large deviation theory we have promoted previously [28]. Further, with the specific choices of the input distributions ψ(τ) and f(χ) we can derive large deviations theory in a straight forward way. The price that we pay is that we do not consider here the theory in its full generality, since we stick to exactly solvable models. The theory of large deviations of stochastic models, for example random walks, has been investigated [32,33,34,35,36,37,38,39,40,41,42], however as we show the fluctuations of the number of jumps, that is a feature of CTRW, is a key issue for appearance of exponential tails.

### 2.1. Displacement Follows Gaussian Distribution

We start the analysis with the simplest choice of waiting times and jump lengths distributions. The jump length PDF is assumed to be Gaussian with zero mean and variance σ=1, i.e., $f\left(\chi \right)=\exp\left(-\chi^{2}/2\right)/\sqrt{2\pi }$. It then follows that P(x|n) is Gaussian as well. The waiting times are exponentially distributed ψ(τ)=exp(−τ) for τ>0. Hence the mean waiting time is 〈τ〉=1 and $Q_{t}\left(n\right)$ obeys Poisson statistics. The density of spreading particles all starting on the origin is therefore
(3)$$\begin{array}{cc} P(x,t)= & \sum_{n=0}^{\infty }\frac{t^{n}\exp\left(-t\right)}{n!}\frac{\exp(-x^{2}/2n)}{\sqrt{2\pi n}} \end{array}$$
and in this sense we have an exactly solvable model valid for any time and *x*. Here the term n=0 contains a delta function on the origin, which of course is not plotted in the figures we present below. This describes particles not moving at all.

To analyze this sum in the large |x| limit we use Cramér-Daniels approach [43] to large deviations. Essentially this is a saddle point method, exploiting the fact that one may relate the cumulant generating function of P(x,t) and the large deviations in the system. For that let us take the Fourier transform of Equation (Equation 3) and then we have
(4)$$\tilde{P}\left(k,t\right)=\sum_{n=0}^{\infty }\frac{t^{n}\exp\left(-t\right)}{n!}{\tilde{f}}^{n}\left(k\right),$$
where we used the Fourier transform of f(χ) denoted $\tilde{f}\left(k\right)=\exp\left(-k^{2}/2\right)$. We have exploited the fact that the jump lengths are independent and identically (IID) hence from convolution theorem of Fourier transform we have the expression $\tilde{P}\left(k|n\right)={\tilde{f}}^{n}\left(k\right)$. The moment generating function is $\langle \exp\left(ux\right)\rangle =\int_{-\infty }^{\infty }P\left(x,t\right)\exp\left(ux\right)dx$. Replacing k→−iu and summing the series
(5)$$\langle \exp\left(ux\right)\rangle =\exp\left[-t(1-e^{u^{2}/2})\right].$$
Then the cumulant generating function is simply the log of the above expression, we denote it
(6)$$K\left(u\right)=\ln\left[\langle \exp(ux)\rangle \right]=-t\left(1-e^{u^{2}/2}\right).$$
Now we are interested in the large |x| limit of P(x,t), which by definition is the inverse Fourier transform $\tilde{P}\left(k,t\right)$. Instead of integrating over *k* one may switch to integration over *u* and using the saddle point method valid for the large |x| one then finds a well known large deviation formula [43]
(7)$$P\left(x,t\right)∼\frac{1}{\sqrt{2\pi K^{\prime \prime }\left(\hat{u}\right)}}\exp\left[K\left(\hat{u}\right)-\hat{u}x\right].$$
Here $\hat{u}$ is given by the solution $K^{\prime }\left(u\right)=x$. While in this section we are treating a special choice of jump length and waiting time, this tool will serve us all along the paper. One should notice that a cumulant generating function does not always exist. For example if f(χ) decays as a power law for large |x|, the moments of the process will diverge. From here we see the first condition of the theory to hold in generality: we assume that cumulant generating function of f(χ) and P(x,t) exist, so the decay of f(χ) is faster than exponential [44].

For the Gaussian choice of f(χ), to find $\hat{u}$ we have
(8)$$K^{\prime }\left(\hat{u}\right)=t\hat{u}\exp\left(\frac{{\hat{u}}^{2}}{2}\right)=x.$$
The solution of this equation is given by the Lambert $W_{0}$ function [45] where the subscript zero denotes the branch of this function. The latter is the solution of the equation yexp(y)=z given by $y=W_{0}\left(z\right)$ if z≥0. Hence we get
(9)$$\hat{u}=\pm \sqrt{W_{0}\left[{\left(\frac{x}{t}\right)}^{2}\right]},$$
where the sign of $\hat{u}$ is dictated by the sign of *x*. Using Equation (Equation 7) we then find
(10)$$P\left(x,t\right)∼\frac{\exp\left(-t-\left|x\right|\left(\sqrt{W_{0}\left[\frac{x^{2}}{t^{2}}\right]}-\frac{1}{\sqrt{W_{0}\left[\frac{x^{2}}{t^{2}}\right]}}\right)\right)}{\sqrt{2\pi K^{\prime \prime }\left(\hat{u}\right)}}$$
with
$$K^{\prime \prime }\left(\hat{u}\right)=\left|x\right|\left(\sqrt{W_{0}\left[{\left(x/t\right)}^{2}\right]}+\frac{1}{\sqrt{W_{0}\left[{\left(x/t\right)}^{2}\right]}}\right).$$
This is plotted in Figure 1. Using the large and small *z* limits of the Lambert function, i.e., $W_{0}\left(z^{2}\right)∼2\ln\left(z\right)$ when z>>1 and $W_{0}\left(z^{2}\right)∼z^{2}$ for |z|<<1, we find the following two limits:(11)$$P\left(x,t\right)≃\exp\left[-x\sqrt{2\ln(x/t)}\right]$$
valid when x/t>>1, and
(12)$$P\left(x,t\right)≃\exp\left[-x^{2}/2t\right]$$
in the opposite limit when |x|/t→0. The first is the mentioned exponential decay, and it includes what we call the ln(|x|) correction. As far as we know, in the experimental literature this ln(|x|) is not reported. However this fact is not surprising as the ln(|x|) is clearly a slowly varying function and hence difficult to detect in reality. We note that the log correction implies that the decay is in reality slightly faster than exponential, implying the existence of the cumulant generating function. The second case is the well known Gaussian behavior found in the centre of the packet.

We may further formulate our results using two very much related approaches, both consider the limits x→∞ and t→∞, and their ratio x/t=l is kept fixed. Using Equation (Equation 10) the first limiting law is
(13)$$\lim_{x\to \infty }-\ln\left(P\left(x,t\right)\right)/x=I_{x}\left(l\right)$$
with $I_{x}\left(l\right)=l^{-1}-1/\sqrt{W_{0}\left(l^{2}\right)}+\sqrt{W_{0}\left(l^{2}\right)}$ and the second refers to
(14)$$\lim_{t\to \infty }-\ln\left(P\left(x,t\right)\right)/t=I_{t}\left(l\right)$$
with $I_{t}\left(l\right)=lI_{x}\left(l\right)$. The functions $I_{t}\left(l\right)$ and $I_{x}\left(l\right)$ are called the rate functions [34,46,47]. All these approaches are essentially identical. Most experimentalists in the field plot P(x,t) versus *x* on a semi-log scale, and for that aim Equation (Equation 10) including the prefactor $1/\sqrt{2\pi K^{\prime \prime }\left(\hat{u}\right)}$ is useful. Mathematicians influenced by long time (large *x*) ideas, and the large deviation literature, would possibly find it natural to use $I_{t}\left(l\right)$ ($I_{x}\left(t\right)$) respectively. In demonstration below we plot the exact solution for finite times, using both approaches; see Figure 1, Figure 2 and Figure 3.

We now present a simple explanation for the exponential tail using a lower bound. Clearly
$$P\left(x,t\right)\geq \frac{\exp(-\frac{x^{2}}{2n^{*}})}{\sqrt{2\pi n^{*}}}\frac{t^{n^{*}}\exp\left(-t\right)}{(n^{*})!}$$
and here $n^{*}$ is the value of *n* for which the summand in Equation (Equation 3) is the maximum. To find $n^{*}$, we use Stirling’s approximation, i.e., $n!∼\sqrt{2\pi n}{\left(n/e\right)}^{n}$, and
(15)$$\frac{d}{dn}\exp(-t+n\ln\left(t\right)-\frac{x^{2}}{2n}-\ln(\sqrt{2\pi n}{\left(\frac{n}{e}\right)}^{n}))=0.$$
The solution of the above equation is $n^{*}∼\left|x\right|/\sqrt{W_{0}\left(x^{2}/t^{2}\right)}$. As expected, in the limit |x|/t→∞, we find the exponential decay again
(16)$$P\left(x,t\right)\geq \exp\left(-t-\left|x\right|\sqrt{\frac{\ln(\frac{\left|x\right|}{t})}{2}}+\left|x\right|\frac{\ln(\frac{te}{\left|x\right|}\sqrt{2\ln(\frac{\left|x\right|}{t}})}{\sqrt{2\ln(|x|/t)}}\right),$$
where we used the asymptotic behavior of the Lambert function, i.e., $W_{0}\left(|x|\right)∼\ln\left(|x|\right)$ with |x|→∞. The claim in [28] is much more general as a similar behavior is found for a large class of waiting times and jump lengths PDFs, see details below.

### 2.2. Displacement Drawn from the Discrete PDF

We now investigate a second example, which further demonstrates the exponential far tails of P(x,t). Consider a CTRW on a one dimensional lattice, with exponential waiting times. Hence f(χ)=[δ(χ−1)+δ(χ+1)]/2 and ψ(τ)=exp(−τ). The Fourier transform of the jump length distribution is $\tilde{f}\left(k\right)=\cos\left(k\right)$ and the moment generating function is 〈exp(uχ)〉=cosh(u), where cosh(u) is hyperbolic cosine function. Here P(x|n) is the well known Binomial distribution
(17)$$P\left(x|n\right)=\frac{1}{2^{n}}\frac{n!}{x!(n-x)!},\;\;\;\;x\leq n,$$
where *x* is an integer since the random walker is on a lattice with a unit spacing. Notice that when *n* is odd/even the same holds for *x*. The CTRW probability is then
(18)$$P\left(x,t\right)=\sum_{n=0}^{\infty }\frac{e^{-t}t^{n}}{n!}P\left(x|n\right).$$
Switching to the Fourier space from the convolution theorem, the Fourier transform of P(x|n) is ${\tilde{f}}^{n}\left(k\right)$. Then after summation it is easy to find the moment generating function M(u)=〈exp(ux)〉=exp[−t(1−cosh(u))]. We then use, as before, the cumulant generating function K(u)=−t[1−cosh(u)]. We now need to find the solution of $K^{{}^{\prime }}\left(u\right)=x$ which is given by $\hat{u}=\sinh^{-1}\left(x/t\right)=\ln\left(x/t+\sqrt{1+x^{2}/t^{2}}\right)$, then using Equation (Equation 7) we get
(19)$$P\left(x,t\right)∼\frac{\exp\left[-x\sinh^{-1}\left(\frac{x}{t}\right)-t+t\sqrt{1+\frac{x^{2}}{t^{2}}}\right]}{\sqrt{2\pi K^{\prime \prime }\left(\hat{u}\right)}}$$
with
$$K^{\prime \prime }\left(\hat{u}\right)=\sqrt{x^{2}+t^{2}}.$$
Equation (Equation 19) provides for small x/t the standard Gaussian behavior, $P\left(x,t\right)≃\exp\left[-x^{2}/2t\right]$. In the opposite limit of large *x*
(20)$$P\left(x,t\right)≃\exp\left[-x\ln\left(\frac{2x}{t}\right)\right],$$
which demonstrates the exponential decay. We may also write this solution like
(21)$$P\left(x,t\right)∼\frac{1}{\sqrt{2\pi K^{\prime \prime }\left(\hat{u}\right)}}\exp\left[-tI_{t}\left(\frac{x}{t}\right)\right]$$
with the rate function
(22)$$I_{t}\left(l\right)=1+l\sinh^{-1}\left(l\right)-\sqrt{1+l^{2}}.$$
Notice that in Equation (Equation 17) P(x|n)=0 for x>n since the particle walking on a lattice cannot reach a distance larger than the number of steps. Thus for standard random walks, with a fixed number of jumps, the exponential tails are not a generic feature. It implies that the fluctuations of the number of steps are crucial for the observation of exponential tails, like those in Equation (Equation 20).

## 3. The Exponential Tails for the Erlang Waiting Time PDF

So far we exposed the exponential tails of P(x,t) assuming that the statistics of the numbers of jumps is Poissonian. To advance the theory further we now consider $Q_{t}\left(n\right)$ for the Erlang PDF of waiting times. For the Erlang distribution, the variance of the waiting time is finite. In the long time limit, from the central limit theorem *n* follows well known Gaussian distribution [31,48]. This indicates that the spreading of the particles obeys Gaussian distribution for the central part of the distribution of the position. For more details see Appendix A. For the tails, large number of jumps *n* is the cause for large displacement from the origin of the random walker. Hence the limit *n* large and *t* fixed, or both these observables large while their ratio is finite is of interest.

### 3.1. The Exponential Tail of the Number of Events

In this regard, two of us have found a general solution of the problem [28]. The large *n* limit of $Q_{t}\left(n\right)$ is controlled by the small τ limit of the waiting time PDF, since to have many jumps within a finite fixed time *t*, the time intervals between jumps must be made small. The main requirement we use here is that ψ(τ) is analytical in the vicinity of τ→0 and hence
(23)$$\psi \left(\tau \right)∼C_{A}\tau^{A}+C_{A+1}\tau^{A+1}.$$
Here *A* is a non negative integer, and $C_{A}$ and $C_{A+1}$ are coefficients. Then in the large *n* limit, and *t* fixed we have [28]
(24)$$Q_{t}\left(n\right)∼\frac{{\left\{{\left[C_{A}\Gamma \left(1+A\right)\right]}^{1/(1+A)}t\right\}}^{n(1+A)}}{\Gamma \left[n(1+A)+1\right]}\exp\left(t\frac{C_{A+1}}{C_{A}}\right).$$
Beyond *A*, $C_{A}$ and $C_{A+1}$ other properties of ψ(τ) are irrelevant. Below, e.g., in Figure 11, we call Equation (Equation 24) the Q large-*n* formula which denotes the limiting law of $Q_{t}\left(n\right)$ for n→∞.

We now consider an example, the Erlang distribution [49]
(25)$$\psi \left(\tau \right)=\frac{\tau^{m-1}e^{-\tau }}{(m-1)!},$$
where *m* is an integer and the mean 〈τ〉=m, see Figure 4. Note that Equation (Equation 25) is a special case of the well known Gamma distribution $\phi \left(\tau \right)=\beta^{\alpha }\tau^{\alpha -1}\exp\left(-\beta \tau \right)/\Gamma \left(\alpha \right)$ with τ,α,β>0. Here exponential waiting times are recovered when m=1. The Laplace transform of ψ(τ) is
(26)$$\hat{\psi }\left(s\right)=\frac{1}{{\left(1+s\right)}^{m}}.$$
Thus, the Erlang PDF of order *m* is the *m* fold convolution of the exponential PDF. As already mentioned, a well known formula Equation (Equation 2) yields the Laplace transform t→s of $Q_{t}\left(n\right)$. Inserting Equation (Equation 26) and using the Laplace pairs
(27)$$\frac{1}{s{\left(1+s\right)}^{nm}}↔1-\frac{\Gamma \left(nm,t\right)}{\Gamma (nm)},$$
where $\Gamma \left(a,z\right)=\int_{z}^{\infty }\tau^{a-1}\exp\left(-\tau \right)d\tau$ is the upper incomplete Gamma function [50], we find
(28)$$Q_{t}\left(n\right)=\frac{\Gamma \left(mn+m,t\right)}{\Gamma \left(nm+m\right)}-\frac{\Gamma \left(nm,t\right)}{\Gamma (nm)},\;\;\;\;\;n\geq 1,$$
and
(29)$$Q_{t}\left(0\right)=\Gamma \left(m,t\right)/\Gamma \left(m\right).$$
Using the identity valid for a positive integer *m*
(30)$$\Gamma \left(m,y\right)=\left(m-1\right)!e^{-y}\sum_{j=0}^{m-1}\frac{y^{j}}{j!},$$
we get
(31)$$Q_{t}\left(n\right)=e^{-t}\sum_{j=nm}^{nm+m-1}\frac{t^{j}}{j!},\;\;\;\;\;\;\;n\geq 1.$$
We will soon use this expression to find an exact solution of the CTRW with the Erlang PDF of waiting times.

From the above theorem, and the definition of the Erlang PDF we have for this example A=m−1, $C_{A}=1/\left(m-1\right)!$ and $C_{A+1}=-C_{A}$. Hence according to Equation (Equation 24), we have
(32)$$Q_{t}\left(n\right)∼e^{-t}\frac{t^{nm}}{(nm)!}.$$
As expected this is the same as the leading term in the expansion Equation (Equation 31), when t/nm≪1. We now find the rate function of this example. Using the Stirling approximation $n!∼\sqrt{2\pi n}{\left(n/e\right)}^{n}$, Equation (Equation 31) reduces to
(33)$$Q_{t}\left(n\right)∼\frac{H(t,l)}{{\left(2\pi mn\right)}^{1/2}}\exp\left[-mnI_{n}\left(l\right)\right],$$
where $I_{n}\left(l\right)=-\ln\left(l\right)+l-1$, l=t/nm and
(34)$$\begin{array}{cc} H(t,l) & =1+\frac{1}{\frac{1}{l}+\frac{1}{t}}+\frac{1}{\left(\frac{1}{l}+\frac{1}{t}\right)\left(\frac{1}{l}+\frac{2}{t}\right)}+\cdots +\frac{1}{\left(\frac{1}{l}+\frac{1}{t}\right)\cdots \left(\frac{1}{l}+\frac{m-1}{t}\right)}. \end{array}$$
For l=t/nm→0, i.e., n→∞ and *m* is fixed, the asymptotic behavior of Equation (Equation 34) follows
(35)$$\begin{array}{cc} H(t,l) & ∼1+l+l^{2}+\cdots +l^{m-1}=\frac{1-l^{m-1}}{1-l}. \end{array}$$
This result is verified in Figure 5. From Equation (Equation 33), the rate function becomes
(36)$$\lim_{n\to \infty }\frac{-\ln Q_{t}\left(n\right)}{mn}=I_{n}\left(l\right),$$
where the limit is valid when *l* is fixed so here *t* is made large. See the red solid line in Figure 6. Finally we consider also
(37)$$\lim_{t\to \infty }\frac{-\ln Q_{t}\left(n\right)}{t}=\bar{l}\ln\left(\bar{l}\right)-\bar{l}+1$$
and here the limit is taken such that $\bar{l}=mn/t$ remains fixed.

**Remark** **1.**
*As mentioned, we use in our plots two graphical representations of the results, the distribution of observables of interest and the rate function. At least to the naked eye we see that in Figure 1, Figure 3 and Figure 5 the results converge already for a relatively short time, e.g., t = 2, while in Figure 2 and Figure 6, we see that the convergence is achieved for much larger times, say t=100. This discrepancy dissolves if all the prefactors like $\ln\left(2K^{\prime \prime }\left(\hat{u}\right)\right)$ are included, and not only the rate function. The rate function formalism is a limit where we take say x→∞ or n→∞. Taking the log of the distribution, be it P(x,t) or $Q_{t}\left(n\right)$ and dividing by a large number we get rid of the prefactors. Hence the two representations are of course not identical.*


### 3.2. The Far Tails of the Positional PDF

For a Gaussian PDF of jump lengths and Erlang’s waiting times we obtain a formal solution from Equations (Equation 1) and (Equation 31)
(38)$$\begin{array}{cc} P(x,t)= & \exp\left(-t\right)\sum_{n=1}^{\infty }\sum_{j=nm}^{nm+m-1}\frac{t^{j}}{j!}\frac{\exp(-\frac{x^{2}}{2n})}{\sqrt{2\pi n}}+\frac{\Gamma (m,t)}{\Gamma (m)}\delta \left(x\right). \end{array}$$
Similar to the previous examples, this function is easy to represent graphically for any reasonable *x* and *t* with a program like MATHEMATICA. Here our goal is to find the large |x| behavior.

As briefly mentioned in the introduction, it was shown previously that P(x,t) decays exponentially with the position *x*, more correctly like |x| multiplied by a slowly varying function. We briefly outline the main result in [28]. Using Cramérs theorem [34] it was shown that under certain conditions, in particular that the PDF of jump length decays faster than exponential decay, $P\left(x|n\right)∼\exp(-n{\left(x/\delta n\right)}^{\beta })$ with β>1 and |x|/n→∞. Here we use Gaussian statistics for the jump lengths so we have $\delta =\sqrt{2}$ and β=2. Then for large *x* the main result in [28] reads
(39)$$P\left(x,t\right)≃\exp\left(-t\left\{\frac{\left|x\right|}{t}Z\left(\frac{\left|x\right|}{t}\right)-\frac{C_{A+1}}{C_{A}}\right\}\right),$$
where
(40)$$Z\left(y\right)=\frac{BW_{0}\left[g_{1}y^{\beta }\right]-g_{0}\left(A+1\right)}{W_{0}{\left[g_{1}y^{\beta }\right]}^{1/\beta }}$$
with
(41)$$B=\frac{g_{0}\left(A+1\right)}{\beta }+\frac{{\left(g_{0}\right)}^{1-\beta }}{\delta^{\beta }},$$
$g_{0}={\left(\beta \left(\beta -1\right)/\left(A+1\right)\right)}^{1/\beta }/\delta$ and $g_{1}={\left[g_{0}\left(A+1\right)/{\left(C_{A}\Gamma \left(A+1\right)\right)}^{1/(1+A)}\right]}^{\beta }$. Recall that the Lambert function $W_{0}\left(y\right)$ is a monotonically increasing function of *y*. Further $W_{0}\left(|y|\right)∼\ln\left(|y|\right)$ for large *y*. Hence from Equation (Equation 39) we learn that P(x,t) decays exponentially (neglecting log terms) as mentioned. This result establishes that the exponential decay of the PDF P(x,t) is a universal feature of CTRW, in the same spirit as experimental and numerical evidence demonstrates in many examples.

To appreciate this result and further test it, let us derive it with our example (in this case the proof is somewhat easier as compared to the general case). When x→∞ and *t* fixed, say *t* of order of c〈τ〉 and c=1 or c=137 etc, the terms contributing to the infinite sum giving P(x,t) are those with a large *n*. Physically and mathematically this is obvious, as to reach large *x* in a finite time we need many jumps since the average of displacement is finite. We therefore use large *n* approximation of $Q_{t}\left(n\right)$ Equation (Equation 31) and then
(42)$$P\left(x,t\right)≃\sum_{n=1}^{\infty }\frac{\exp\left(-t\right)t^{nm}}{(nm)!}\frac{\exp(-\frac{x^{2}}{2n})}{\sqrt{2\pi n}}.$$
Only large *n* terms contribute, hence including small *n* in the summation is not a problem, since these terms are of order $\exp(-x^{2})$ while |x|→∞. We now switch to Fourier space, and find
(43)$$\tilde{P}\left(k,t\right)≃\exp\left(-t\right)S_{m}\left[t^{m}\exp\left(-k^{2}/2\right)\right]$$
with the sum $S_{m}\left(y\right)=\sum_{n=0}^{\infty }y^{n}/\left(nm\right)!.$ The next step is to find the asymptotic behavior of $S_{m}\left(y\right)$ for large *y*. In order to do so, let us consider some special cases. Hence for m=1, $S_{1}\left(y\right)=\exp\left(y\right)$, and similarly $S_{2}\left(y\right)=\cosh\left(\sqrt{y}\right)∼\exp\left(\sqrt{y}\right)/2$,
(44)$$\begin{array}{cc} S_{3}\left(y\right) & =\frac{1}{3}e^{-\frac{\sqrt[{3}]{y}}{2}}\left(e^{\frac{3\sqrt[{3}]{y}}{2}}+2\cos\left(\frac{1}{2}\sqrt{3}\sqrt[{3}]{y}\right)\right)\\ \; & ∼\frac{1}{3}\exp\left(-y^{1/3}\right), \end{array}$$
etc. For a general *m*, we write the infinite sum as an integral
(45)$$\begin{array}{cc} S_{m}\left(y\right) & \to \int_{0}^{\infty }\frac{y^{n}}{(nm)!}dn\\ \; & =\frac{1}{m}\int_{0}^{\infty }\frac{{\left(y^{1/m}\right)}^{j}}{(j)!}dj\\ \; & ∼\frac{1}{m}\exp\left(y^{1/m}\right). \end{array}$$
Note that Equation (Equation 45) can also be obtained directly by using the asymptotic behavior of the Mittag-Leffler function $E_{\alpha ,\beta }\left(y\right)=\sum_{j=0}^{\infty }y^{j}/\Gamma \left(\alpha j+\beta \right)$ since $S_{m}\left(y\right)=E_{m,1}\left(y\right)$ [51]. Below, we will use the above equation to calculate the far tails of the distribution of the position. As usual switching to k=−iu we obtain the moment generating function, and taking the log, we get the cumulant generating function
(46)$$K\left(u\right)≃-t+t\exp(u^{2}/\left(2m\right)).$$
This is valid for large *u* since that limit is the relevant one for the calculations of P(x,t) for large *x*, similar to our previous examples. Here we neglected a ln(m) term that is negligible in the limit under study. Using the standard large deviation technique Equation (Equation 7), we find the solution of $K^{{}^{\prime }}\left(u\right)=x$ denoted $\hat{u}$ and this solves the equation
(47)$$\hat{u}\exp\left(\frac{{\left(\hat{u}\right)}^{2}}{2m}\right)=\frac{mx}{t},$$
and hence we find yet again the Lambert type of solution
(48)$$\hat{u}=\sqrt{mW_{0}\left[m\left(\frac{x^{2}}{t^{2}}\right)\right]}$$
for x>0 otherwise the right hand side of Equation (Equation 48) has a negative sign. If m=1, as expected, we relax to Equation (Equation 9). Using the large deviation formula Equation (Equation 7), we find
(49)$$P\left(x,t\right)∼\frac{\exp\left(-\sqrt{m}\left|x\right|\sqrt{W_{0}\left(\frac{mx^{2}}{t^{2}}\right)}+\frac{\sqrt{m}\left|x\right|}{\sqrt{W_{0}\left(\frac{mx^{2}}{t^{2}}\right)}}-t\right)}{\sqrt{\left|\left|x\right|2\pi m^{-1/2}\left(\sqrt{W_{0}\left(\frac{mx^{2}}{t^{2}}\right)}+\frac{1}{\sqrt{W_{0}\left(\frac{mx^{2}}{t^{2}}\right)}}\right)\right|}}.$$
This solution and the more general one Equation (Equation 39) are of course in agreement. As shown in Figure 7, the far tails of the distribution of the position follow exponential decay predicted by Equation (Equation 49) since as mentioned $W_{0}\left(z\right)∼\ln\left(z\right)$. But as shown in Figure 7 the rate of convergence is slow, for example for m=3.

To check better the convergence issue, we consider the rate function. Based on Equation (Equation 49), the rate function with respect to the position becomes
(50)$$\lim_{t\to \infty }-\frac{\ln(P(x,t))}{\left|x\right|}=I_{x}\left(l\right)$$
with l=x/(t/〈τ〉) and
(51)$$I_{x}\left(l\right)=\sqrt{m}\left(W_{0}\left(l^{2}/m\right)-\frac{1}{W_{0}\left(l^{2}/m\right)}\right)+\frac{m}{\left|l\right|}.$$

It can be seen in Figure 8 that with the increase of observation time *t*, the left-hand of Equation (Equation 50) tends to $I_{x}\left(l\right)$ (Equation (Equation 51)) slowly. One contribution to this slow convergence effect, is that *t* is not very large, the prefactor of $\exp(-|x|I_{x}\left(l\right))$, i.e., $1/\sqrt{2\pi K^{\prime \prime }\left(\hat{u}\right)}$, is of importance.

As expected, when $mx^{2}/t^{2}\to 0$, we get the Gaussian distribution
(52)$$P\left(x,t\right)≃\exp\left(-\frac{x^{2}m}{2t}\right)$$
and since 〈τ〉=m from Equation (Equation 25) we have $P\left(x,\bar{t}\right)≃\exp\left(-x^{2}/2\bar{t}\right)$ with $\bar{t}=t/\langle \tau \rangle$. Recall that $W_{0}\left(y\right)∼\ln\left(y\right)$ for y≫1, hence when $mx^{2}/t^{2}\gg 1$ we may approximate Equation (Equation 49) with
(53)$$P\left(x,t\right)≃\exp\left(-\sqrt{m}\left|x\right|\sqrt{\ln\left(\frac{mx^{2}}{t^{2}}\right)}-t\right).$$
Thus using the rescaled time $\bar{t}=t/\langle \tau \rangle$ we have
$$P\left(x,t\right)≃\exp\left(-\sqrt{m}\left|x\right|\ln\left[\frac{x^{2}}{m{\bar{t}}^{2}}\right]-t\right),$$
implying that as we increase *m* for fixed $\bar{t}$ we get a suppression of the exponential tails.

As we increase *m* the condition on $x\gg \sqrt{m}\bar{t}$ implies that we may observe the nearly exponential decay of the packet but only for a very large *x*, or only for very short times. This is because when we increase *m*, the waiting time exhibits a strong anti-bunching effect. Namely, to find a large *x* we need to have a large fluctuations of *n*. For example if the average of 〈n(t)〉∼2 we may still see realizations with many jumps, leading to exponential decay in the far tails of *x*. However, this is less likely for large *m* if compared with small *m*; see Figure 4. This is because when *m* is large we have vanishing fluctuations of *n*. To quantify this we may use a tool from quantum optic, namely the Mandel *Q* parameter [52] defined as
(54)$$Q=\frac{\langle n^{2}\rangle -{\langle n\rangle }^{2}}{\langle n\rangle }-1.$$
The case Q<0 is called sub-Poissonian (anti-bunching) and Q>0 sup-Poissonian. If Q=−1, we have no fluctuations of *n* at all. For the Erlang PDF, and in the long time limit Q=−1+1/m. Namely, when m→∞ the fluctuations of *n* vanish (see Figure 9), and hence with it the effect of exponential tails of P(x,t). This is easy to see since in the absence of fluctuations of *n*, the number of jumps is fixed and since we have Gaussian jump lengths, the total displacement will be Gaussian and not exponential. Thus we see that as we increase *m*, namely make the process more anti-bunched we see less exponential tails. Anti-bunching means the effective repulsion of the dots on the time axis on which jump events takes place. And this is controlled by the small τ behavior of ψ(τ). The anti-bunching *Q* reduces to a negative value, and hence kills fluctuations of *n* which are the key to the observation of exponential tails of P(x,t).

## 4. Bunching Case of Waiting Time PDF

We further explore the behavior of probability to observe a large number of renewals *n*. For some special distributions of the waiting times it is unwise simply to take the n→∞ limit since the rate of convergence to Equation (Equation 24) is very slow. Here we consider ψ(τ) as a sum of two exponential waiting times PDFs
(55)$$\psi \left(\tau \right)=\frac{1}{2}\left(\frac{1}{a}\exp\left(-\frac{\tau }{a}\right)+\frac{1}{b}\exp\left(-\frac{\tau }{b}\right)\right).$$
We choose a<b while the opposite situation is merely relabeling. The average waiting time is 〈τ〉=(a+b)/2. In what follows we shall fix 〈τ〉=1, so a+b=2 and this means that we have one free parameter say *a* with 0<a<1. We also consider the case where the jump size distribution is Gaussian with a variance equal unity and zero mean. This implies that in the long time limit, independently of the specific choice of *a*, the mean square displacement $\langle x^{2}\rangle ∼t$ and the process is Gaussian. However, for short times the dynamics of P(x,t) is of course *a* sensitive. The question we wish to address: how does *a* control the exponential tails of P(x,t)?

This is related to effect of bunching. Here for short waiting times we have ψ(0)=(1/a+1/b)/2. When a=b=1 we have a Poisson process, while when say a→0, we get a large value of ψ(0). This means that there is an increased probability (compared to Poissonian case) to obtain a jump shortly after a jump event. This is an effect of bunching where the jumps come in groups, and then separated by the relatively large waiting times. Such intermittent behavior is quantified with Mandel *Q* parameter, which for large time is
(56)$$Q=2{\left(a-1\right)}^{2},$$
where we assumed that a+b=2. This is a super Poissonian behavior since Q>0. Note that in the previous example of the Erlang PDF, we had the opposite behavior, i.e., Q<0, since there $\lim_{\tau \to 0}\psi \left(\tau \right)=0$ for any m≠1.

We can obtain $Q_{t}\left(n\right)$ in terms of an integral; see details in Appendix B. Aside $Q_{t}\left(n\right)$ we also find P(x,t), which is compared to the previous theory when x→∞, i.e., Equation (Equation 39); see Figure 10. Here we see that by making *a* smaller, namely making the process more bunched, we get large exponential tails (see Figure 10). Thus bunching makes the observation of the exponential tails effect more readily achievable in experiments. In principle for any PDF of the waiting times (provided ψ(τ) is analytical in the vicinity of τ=0) an exponential decay of P(x,t) is obtained, but this decay can be sometimes achieved for extremely large values of *x*. We also observe that simply including the asymptotic behavior of $Q_{t}\left(n\right)$ in Equation (Equation 24) leads to slow convergence in the a→0 limit. To further understand the effect of bunching we also plot $Q_{t}\left(n\right)$ versus *n*; see Figure 11. Strong bunching in this model, implies a relatively high probability for seeing a large *n* (many short time intervals between jumps since *a* is small). As we see, the decay of $Q_{t}\left(n\right)$ with *n* is relatively slow when bunching is pronounced, and then the appearance of non negligible probability for large *n*, implies that particles can travel large distances, and then the tails posses more statistical weight. In Figure 11 the slow convergence to asymptotic behavior of Equation (Equation 24) is observed in the a→0 limit and the delicate treatment of Appendix B is preferable.

## 5. Conclusions

Following the work of Kob and co-workers we have formalized the problem of nearly exponential decay of P(x,t) using the CTRW framework. Exponential decay is the rule, and it should be considered a natural consequence of large deviation theory. In the long time limit the packet of particles is typically Gaussian, hence the phenomenon can be found/measured for intermediate and short times. To describe the dynamics we have considered a few examples where we may find exact solutions to the problem. This allowed us to find the far tail, obtain the rate function, and compare finite time solutions with asymptotic expressions.

We distinguished between bunching and anti-bunching processes. These are determined by the behavior of ψ(τ) in the vicinity of of τ close to zero. The short time behavior of ψ(τ) determines the statistical behavior of the number of jumps, when the latter is large. And large number of jumps, lead the particle to non-typical large *x*, where the phenomenon is found. Using the example of ψ(τ) expressed as a sum of two exponential waiting times, we show that as we increase the bunching effect, the exponential tails of P(x,t) are more pronounced. Similarly as we increase anti-bunching, by increasing the parameter *m* in the Erlang distribution, the exponential tails are suppressed. Indeed as m→∞ we get the usual random walk where *n* is not fluctuating and then exponential tails cannot be found.

These effects are related to the rather universal behavior of $Q_{t}\left(n\right)$ found for large *n*. As we have demonstrated here, for large *n*
$Q_{t}\left(n\right)$ universally attains exponential decay (with log corrections). This, as mentioned, is valid for any ψ(τ) which is analytic for small τ. Thus the agreement with the experimental observation that finds exponential tails, is a manifestation of the widely applicable CTRW model under study and not merely consequence of a fitting procedure. This universal property of CTRW is the crucial difference from the diffusing diffusivity model [10,18,19,53,54,55,56,57].

When dealing with a regular diffusive motion the only transport parameter that determines the behavior is the diffusive constant *D*. The diffusive constant also determines the position range where the Gaussian behavior is to be expected, i.e., for any *x* such that $\left|x\right|\propto {\left(Dt\right)}^{1/2}$. *D* itself is defined as a ratio of two microscopic quantities, the variance of the jump lengths ($\sigma^{2}$) and the average time between jumps 〈τ〉, i.e., $D=\sigma^{2}/2\langle \tau \rangle$. The tails of the density, investigated in this paper, are clearly not determined by D. When the jumps are distributed according to a Gaussian distribution with the variance $\sigma^{2}$ and the waiting time posses an exponential distribution, with 〈τ〉 as a mean waiting time, the exponential tails are expected for any *x* such that |x|/σ>t/〈τ〉. We see that for this specific case the important quantity is not *D* but rather σ/〈τ〉. While σ and 〈τ〉 are the microscopic parameters that determine the large, and small *x* for a given time *t*, for any other case with no Gaussian jumps or exponential distribution of waiting times other parameters emerge. The average waiting time 〈τ〉 is replaced by $1/{\left(C_{A}\right)}^{1/(A+1)}$ from the expansion of ψ(τ) in Equation (Equation 23) and σ is replaced by δ whose role was exposed in [28], a parameter that determines f(χ) for large |χ| ($f\left(|\chi |\right)∼{\exp(-(|\chi |/\delta )}^{\beta })$ when |χ|→∞). Overall the condition for appearance of exponential tails is modified to $|x|>t{\left(C_{A}\right)}^{1/(A+1)}\delta$ (see Reference [28]). Another thing to notice about this condition is that the presented results are not limited to large times but emerge for any sufficiently large |x|.

As mentioned in Section 3, in this manuscript we assume that the PDF of waiting time ψ(τ) is analytic at the vicinity of τ=0. Clearly not all of the PDFs of waiting time satisfy this property, for example $\psi \left(\tau \right)=\tau^{-3/2}\exp\left[-1/\left(4\tau \right)\right]/2\sqrt{\pi }$. Roughly speaking, for this case the probability of performing many jumps at a finite time is much smaller than the cases studied here since ψ(τ→0)=0. More specifically, for this example the analytical behavior Equation (Equation 23) is not valid, and hence our main results do not apply. This implies that the far tails of distribution of the position decay faster than exponential tails. It would be of interest to investigate this problem, which is left for future work.

Finally, let us mention a few other open problems:In the present case we assumed that the jump process starts at time t=0. This is called an ordinary renewal process. If the processes started long before the measurement, we will have a modification of the PDF of the first waiting time [3,58]. How does this effect the large *x* behavior of P(x,t)? This issue seems important since the phenomenon can be found for relatively short times.We focused on models that in the long time limit converge to Gaussian statistics. What happens if ψ(τ) is fat tailed [59,60] with diverging mean? Our results are certainly valid for this case as well, however we did not explore this in detail.What happens in dimension d>1?What are ideal waiting time PDFs and jump length distributions, where exponential tails are pronounce and if possible maintained for longer times. We showed how this is related to bunching and anti-bunching, however more refined work can help to clarify a better the widely observed behavior.We used CTRW, instead one could use the noisy CTRW model [61]. This adds to the jumps also noise when the particle is waiting for its next jump. Thus noisy CTRW is much more similar to real experiments.Here we considered the decoupled CTRW, where jump lengths and waiting times are uncorrelated. The general framework of CTRW, goes beyond this simplification [62,63].If the jump length PDF is sub-exponential, the far tail of P(x,t) will deviate from what we found here. Most likely the principle of the single big jump [64,65,66] will hold in some form, but the details of the theory are left unknown.Recently Dechant et al. showed how the CTRW picture emerges from an under-damped Langevin description of a particle in a periodic potential [29]. And then showed how this model can be used to analyse dynamics of Cesium atoms in optical lattices. Thus we expect to find also here exponential tails of packets, however influence of the control parameters of this phenomenon such as the depth of the optical potential, the noise etc, are left unknown to us. Similarly, over damped Brownian motion in corrugated channels, a model of biophysical transport, is likely related to CTRW as a coarse grained description. In the former system exponential decay was already explored in Reference [67]. Thus exponential tails are found both via Langevin dynamics and within CTRW, the two approaches are related in some limits.

## Figures and Tables

**Figure 1 entropy-22-00697-f001:**
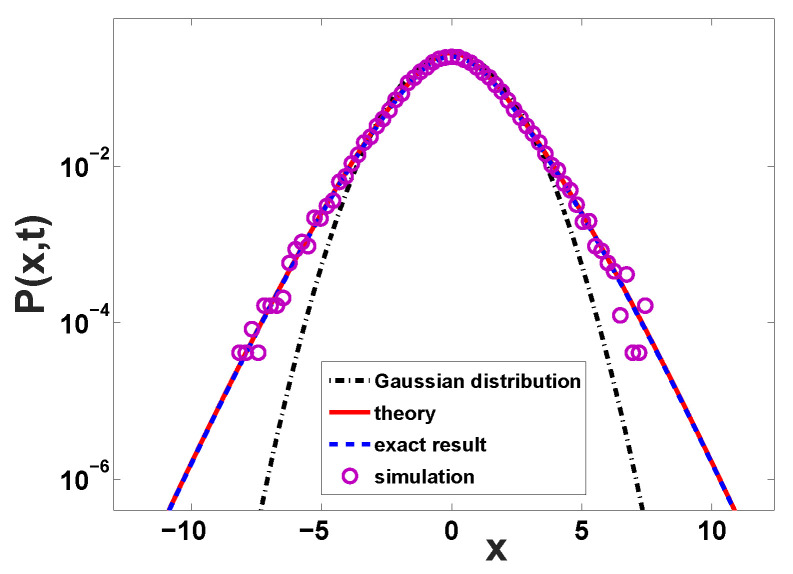
The distribution P(x,t) with exponentially distributed waiting times and Gaussian displacements. The time is t=2 and for simulations we used $5\times 10^{7}$ trajectories. Our theory Equation (Equation 10) performs perfectly, while the Gaussian distribution Equation (Equation 12), black dash-dotted line, completely fails for the far tails of the distribution of the position. Note that the theoretical prediction Equation (Equation 10) works extremely well also for the central part of the distribution. Here the exact result is obtained from Equation (Equation 3).

**Figure 2 entropy-22-00697-f002:**
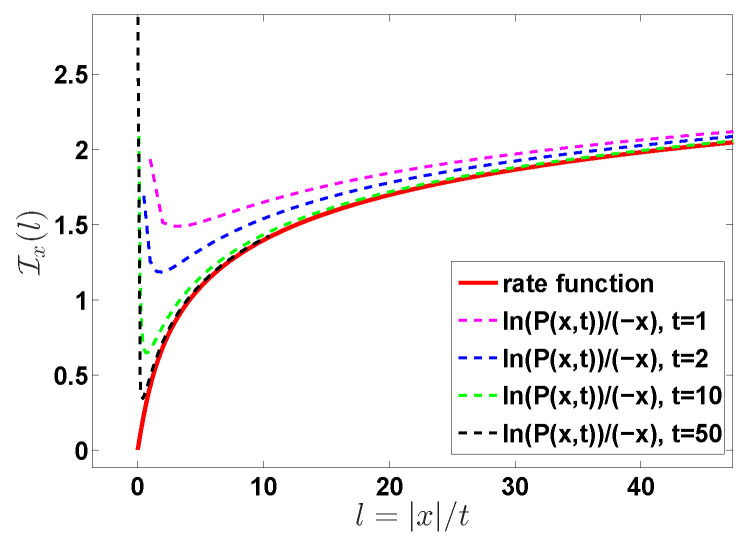
The plot of the rate function $I_{x}\left(l\right)$ with l=|x|/t. The red solid line, predicted by Equation (Equation 13), describes the rate function $I_{x}\left(x/t\right)$ capturing the behavior of large deviations. Note that for large *l* we have $I_{x}\left(l\right)∼\sqrt{2\ln(l)}$ and this means that P(x,t) is decaying exponentially with the distance *x* with a correction. The dashed lines are the plot of ln(P(x,t))/(−|x|) calculated from Equation (Equation 3).

**Figure 3 entropy-22-00697-f003:**
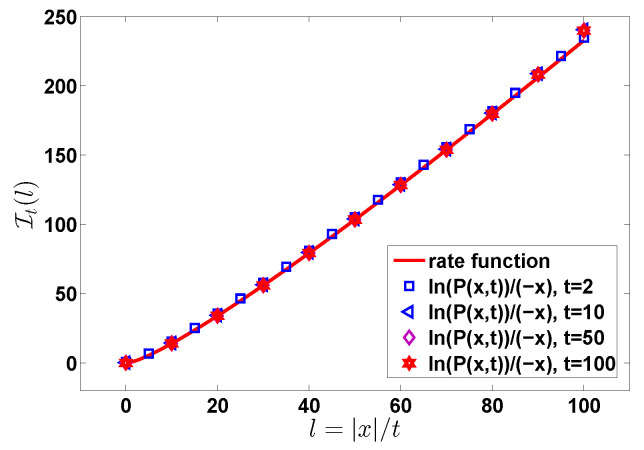
Rate function $I_{t}\left(l\right)$ versus l=|x|/t for different *t*. The red solid line is the (time) rate function Equation (Equation 14). As in Figure 1 and Figure 2 here ψ(τ)=exp(−τ) and $f\left(\chi \right)=\exp\left(-\chi^{2}/2\right)/\sqrt{2\pi }$.

**Figure 4 entropy-22-00697-f004:**
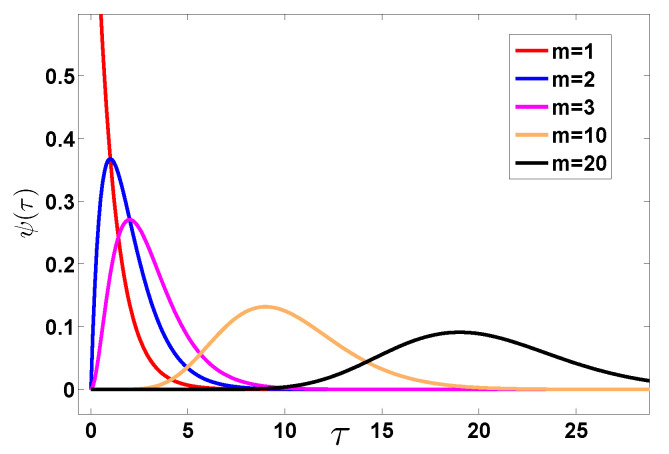
Plot of the Erlang PDF (Equation (Equation 25)) for various *m*. Increasing *m* leads to decreasing probability of obtaining small τ that leads to transition from bunching to anti-bunching [see discussion preceding Equation (Equation 54)].

**Figure 5 entropy-22-00697-f005:**
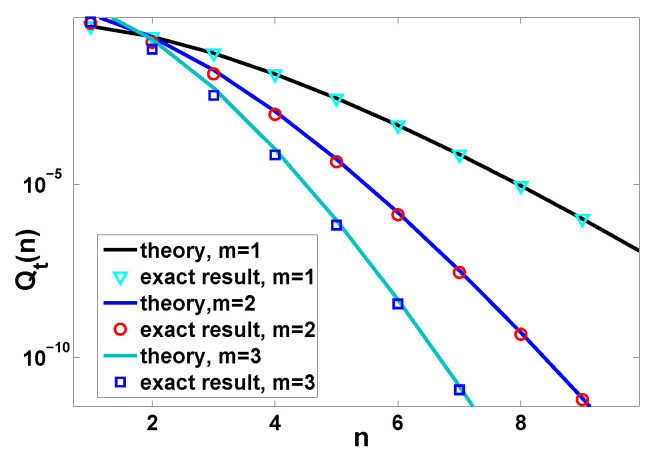
Probability to observe *n* renewals when the waiting time is determined by Equation (Equation 25) for different *m*. The solid lines are theoretical prediction Equation (Equation 33) with H(t,l) obtained from Equation (Equation 35) and the corresponding exact results, plotted by the symbols, are Equation (Equation 28). In our setting, we fix $\bar{t}=t/\langle \tau \rangle =t/m=1$ and change the value of *m*.

**Figure 6 entropy-22-00697-f006:**
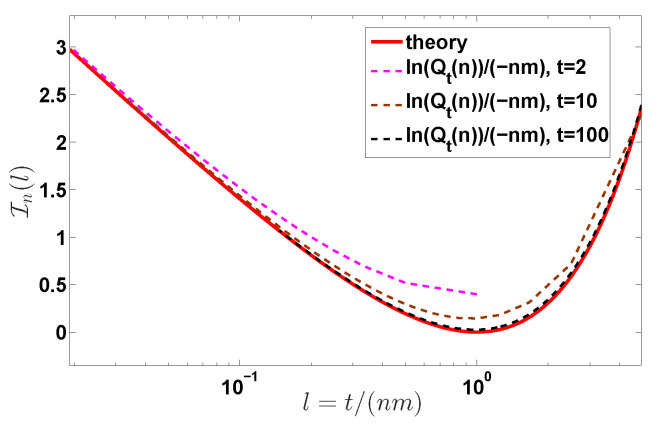
Rate function $I_{n}\left(l\right)$ versus l=t/(nm) for different observation times *t* with m=2. The dashed lines are the plot of the function $\ln(Q_{t}\left(n\right))/\left(-nm\right)$ with $Q_{t}\left(n\right)$ calculated from Equation (Equation 31). Clearly, with the increase of the observation time *t*, the function $\ln(Q_{n}\left(t\right))/\left(-nm\right)$ versus *l* approaches to the rate function $I_{n}\left(l\right)$.

**Figure 7 entropy-22-00697-f007:**
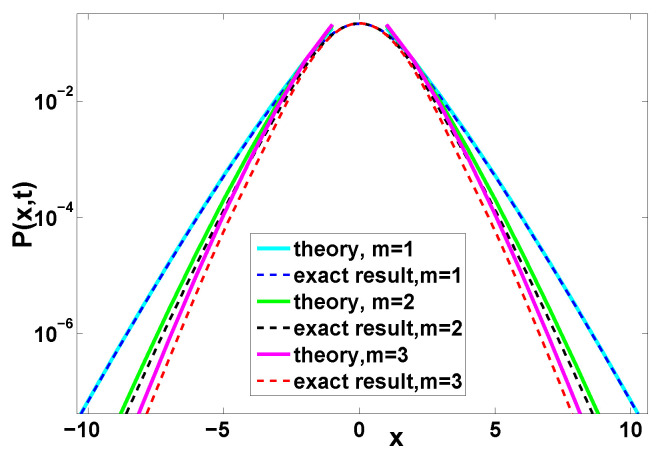
Distribution of the position with the Erlang distribution Equation (Equation 25), where we chose the rescaled time $\bar{t}=1$. Here the solid lines are the theory according to Equation (Equation 49) and the corresponding exact result is obtained from Equation (Equation 1). See Figure 8 for the convergence of the theory.

**Figure 8 entropy-22-00697-f008:**
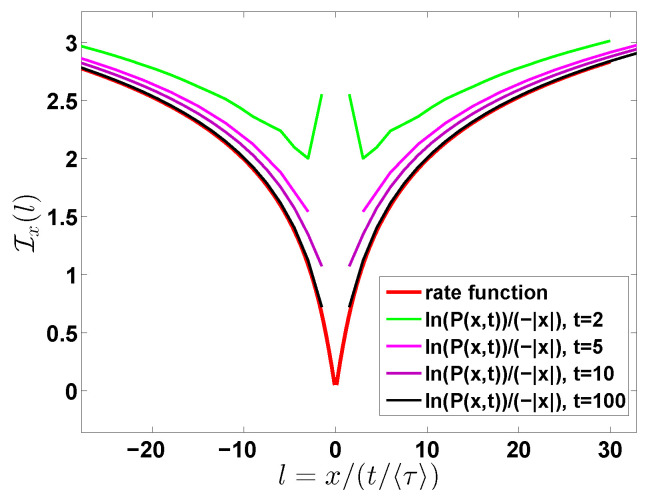
Comparison of analytical prediction Equation (Equation 50) (red line) for $I_{x}\left(l\right)$ with ln(P(x,t))/(−|x|). We show that ln(P(x,t))/(−|x|) obtained from Equation (Equation 38) converges to the rate function Equation (Equation 50) with the growing of the observation time *t*. Here we choose m=3.

**Figure 9 entropy-22-00697-f009:**
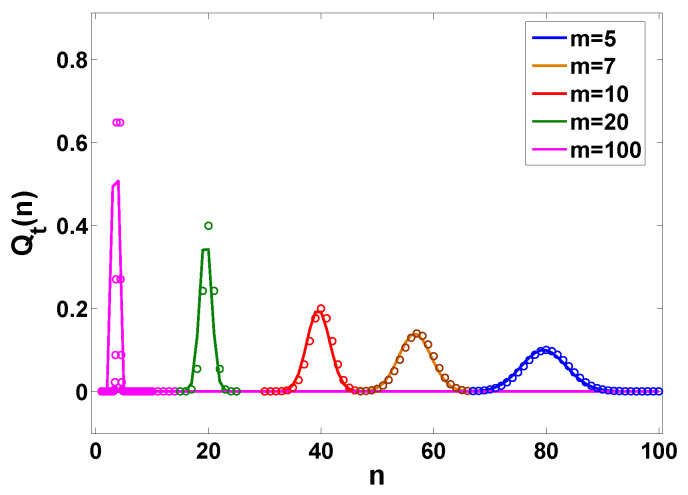
$Q_{t}\left(n\right)$ for the case of Erlang PDF of waiting times, i.e., ψ(τ) is determined by Equation (Equation 25). The solid lines describe Equation (Equation 28) the exact behavior of $Q_{t}\left(n\right)$ for various *m*. Here we choose t=400. When *m* increases the distribution narrows, indicating that fluctuations of *n* disappear. We further plot the central part of $Q_{t}\left(n\right)$ using the symbols according to Gaussian approximation. Here for Gaussian approximation we use $Q_{t}\left(n\right)∼\exp\left(-{\left(n-t/m\right)}^{2}/\left(2t/m^{2}\right)\right)/\sqrt{2\pi t/m^{2}}$ (see Appendix A).

**Figure 10 entropy-22-00697-f010:**
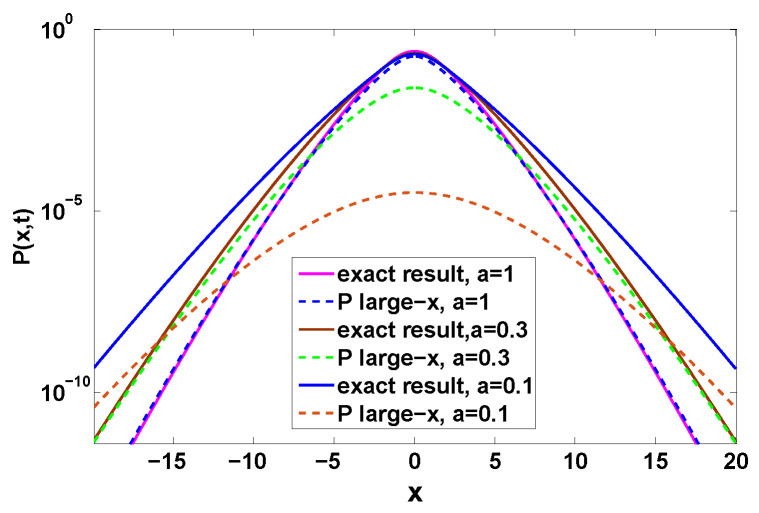
P(x,t) for the case where the jump lengths are Gaussian and ψ(τ) is given by Equation (Equation 55). The solid line describes the exact behavior for various *a*, as provided by Equations (Equation 1) and (Equation 61), while the corresponding large |x| approximations (termed *P* large-*x*) due to Equation (Equation 39) described by the dashed lines. For small *a*, e.g., a=0.1, the convergence of the large |x| approximation is slow and achieved only for very large values of |x|. See Appendix B for a proper discussion of the behavior in this limit.

**Figure 11 entropy-22-00697-f011:**
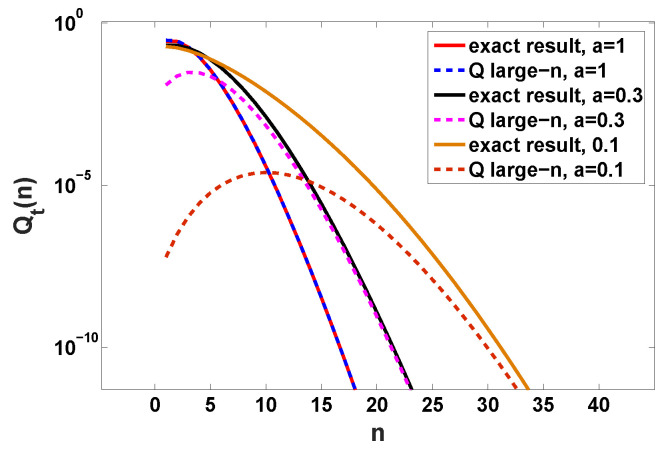
Probability $Q_{t}\left(n\right)$ of observing *n* jump events versus *n* while ψ(τ) is determined by Equation (Equation 55). The measurement time is t=2. The solid lines describes the exact results for various *a*, i.e., Equation (Equation 61), and the dashed lines are the corresponding Q large-*n* approximations, i.e., Equation (Equation 24).

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
