# Peer review of "Large Deviations for Continuous Time Random Walks"

_entropy, 2020, doi:10.3390/e22060697_

Round 1

Reviewer 1 Report

Report on the manuscript "Large deviations for continuous time random walks"

The authors develop and further extend their breakthought contribution
in [27] that explained frequent observations of the exponential decay
of the position PDF, found in numerous experiments. As this decay may
seem contradicting the central limit theorem, many arguments have been
proposed to rationalize this experimental evidence. The large deviation
theory for continuous time random walks, originated in [27] and further
elaborated in the manuscript, presents an elegant and universal mechanism
for exponential tails. More precisely, the authors discuss several
exactly solvable models, which allow them to investigate the range of
validity of their general theory. The manuscript is timely, well written
and appropriate for the Entropy journal. For these reasons, I recommend
its publication after a minor revision.

MINOR POINTS

1) l. 40: "However a second limit is important, which is large positions
(and finite time)."
Although the meaning of this sentence becomes clear after reading few
pages, it is suggested to slightly reformulate it for clarity.

2) Sec. 2. The authors introduce their models by using dimensionless
position x and time t (by setting the mean waiting time to 1
and the displacement variance to 1). In particular, they distinguish
different asymptotic regimes such as, e.g., x/t >> 1. This is a quite
natural choice for a theoretical paper. However, to make the paper
more accessible for non-theoretical community, it is recommended to
discuss somewhere (may be, in conclusion section) the major findings
within conventional units. This discussion will also help to highlight
the main findings. For instance, the common short- and long-time
asymptotics for Brownian motion would correspond to (Dt)^{1/2} << x
and (Dt)^{1/2} >> x, where D is the diffusion coefficient, which is
the only macroscopic parameter of the dynamics. What are the physically
measurable parameters that are relevant in the present work? It seems
that it should be the mean waiting time <\tau> and the variance \sigma^2.
I believe that such a discussion would further increase the impact of
this paper.

3) line after Eq. (13): remove the second closing parenthesis in the
expression for I_x(l).

4) l. 91: As the authors refer to the asymptotic behavior of the
Lambert function, does inequality (16) hold strictly or just
asymptotically? This is just a technical mathematical issue.

5) In Fig. 2, the axes labels are too small and hard to read,
please enlarge.
Same for Fig. 3 (here, the text in the caption is also rather small).
Same for Fig. 6 (here, the text in the caption is also rather small).
Same for Fig. 8 (here, the text in the caption is also rather small).

6) l. 110: as the abbreviation "Qln" is not common, it is
suggested to spell it out explicitly once (Q large-n)?

7) After l. 110: It is suggested to mention that the Erlang
distribution (considered in the paper) is the special case
of the gamma distribution (which is more commonly known).
This can also be an occasion for authors to highlight that
they restrict the shape parameter m to be integer to ensure
that \psi(\tau) is analytic at 0.

8) After Eq. (27), it is suggested to specify that Gamma(a,t)
is the _upper_ incomplete Gamma function.

9) In Eq. (34), what is j ?
Likewise, in Eq. (35), what is j? How the expression is obtained?
It looks as a sum of the geometric sequence but the expression
seems to be wrongly typed.
Please clarify this passage.

10) l. 114: "As mentioned we, use in" -> "As mentioned, we use in"

11) After Eq. (43). The function S_m(y) is the Mittag-Leffler
function, its properties (including Eqs. (44,45)) are well known:
https://en.wikipedia.org/wiki/Mittag-Leffler_function

12) l. 174: "conclusion" -> "Conclusion"

13) Fig. 9: why there are no symbols (circles) on the peak for m=100 ?

14) Caption of Fig. 10: "The solid line describe" -> "The solid lines describe"

15) Caption of Fig. A3: "and and" -> "and"

Author Response

Thanks!

Sincerely,

Wanli Wang

Reviewer 2 Report

The Authors analyze an interesting problem of CTRW and obtain the large deviation description of the propagator. Different waiting time and jump length PDFs are considered. The paper is well written and of current interest for many scholars. At the end of the paper, Authors give a list of related open problems, which will be of interest for investigation in the future. Therefore, the manuscript is worth to be published in the journal Entropy.
